# Hydralazine Sensitizes to the Antifibrotic Effect of 5-Aza-2′-deoxycytidine in Hepatic Stellate Cells

**DOI:** 10.3390/biology9060117

**Published:** 2020-06-03

**Authors:** Kiyoshi Asada, Kosuke Kaji, Shinya Sato, Kenichiro Seki, Naotaka Shimozato, Hideto Kawaratani, Hiroaki Takaya, Yasuhiko Sawada, Keisuke Nakanishi, Masanori Furukawa, Mitsuteru Kitade, Kei Moriya, Tadashi Namisaki, Ryuichi Noguchi, Takemi Akahane, Hitoshi Yoshiji

**Affiliations:** Third Department of Internal Medicine, Nara Medical University, Kashihara, Nara 634-8521, Japan; kajik@naramed-u.ac.jp (K.K.); shinyasato@naramed-u.ac.jp (S.S.); seki@naramed-u.ac.jp (K.S.); shimozato@naramed-u.ac.jp (N.S.); kawara@naramed-u.ac.jp (H.K.); htky@naramed-u.ac.jp (H.T.); yasuhiko@naramed-u.ac.jp (Y.S.); nakanishi@naramed-u.ac.jp (K.N.); furukawa@naramed-u.ac.jp (M.F.); kitadem@naramed-u.ac.jp (M.K.); moriyak@naramed-u.ac.jp (K.M.); tadashin@naramed-u.ac.jp (T.N.); rnoguchi@naramed-u.ac.jp (R.N.); stakemi@naramed-u.ac.jp (T.A.); yoshijih@naramed-u.ac.jp (H.Y.)

**Keywords:** hydralazine, 5-aza-2′-deoxycytidine, DNA methylation, hepatic stellate cell, liver fibrosis

## Abstract

Background: Hepatic stellate cell (HSC) activation is essential for the development of liver fibrosis. Epigenetic machinery, such as DNA methylation, is largely involved in the regulation of gene expression during HSC activation. Although the pharmacological DNA demethylation of HSC using 5-aza-2′-deoxycytidine (5-aza-dC) yielded an antifibrotic effect, this drug has been reported to induce excessive cytotoxicity at a high dose. Hydralazine (HDZ), an antihypertensive agent, also exhibits non-nucleoside demethylating activity. However, the effect of HDZ on HSC activation remains unclear. In this study, we performed a combined treatment with 5-aza-dC and HDZ to obtain an enhanced antifibrotic effect with lower cytotoxicity. Methods: HSC-T6 cells were used as a rat HSC cell line in this study. The cells were cultivated together with 1 µM 5-Aza-dC and/or 10 µg/mL of HDZ, which were refreshed every 24 h until the 96 h treatment ended. Cell proliferation was measured using the WST-1 assay. The mRNA expression levels of peptidylprolyl isomerase A (*Ppia*), an internal control gene, collagen type I alpha 1 (*Cola1*), RAS protein activator like 1 (*Rasal1*), and phosphatase and tensin homolog deleted from chromosome 10 (*Pten*) were analyzed using quantitative reverse transcription polymerase chain reaction. Results: The percentage cell viability with 5-aza-dC, HDZ, and combined treatment vs. the vehicle-only control was 101.4 ± 2.5, 95.2 ± 5.7, and 79.2 ± 0.7 (*p* < 0.01 for 5-aza-dC and *p* < 0.01 for HDZ), respectively, in the 48 h treatment, and 52.4 ± 5.6, 65.9 ± 3.4, and 29.9 ± 1.3 (*p* < 0.01 for 5-aza-dC and *p* < 0.01 for HDZ), respectively, in the 96 h treatment. 5-Aza-dC and the combined treatment markedly decreased *Cola1* mRNA levels. Accordingly, the expression levels of *Rasal1* and *Pten*, which are antifibrotic genes, were increased by treatment after the 5-aza-dC and combined treatments. Moreover, single treatment with HDZ did not affect the expression levels of *Cola1*, *Rasal1*, or *Pten*. These results suggest that HDZ sensitizes to the antifibrotic effect of 5-aza-dC in HSC-T6 cells. The molecular mechanism underlying the sensitization to the antifibrotic effect of 5-aza-dC by HDZ remains to be elucidated. The expression levels of rat equilibrative nucleoside transporter genes (*rEnt1*, *rEnt2*, and *rEnt3*) were not affected by HDZ in this study. Conclusions: Further confirmation using primary HSCs and in vivo animal models is desirable, but combined treatment with 5-aza-dC and HDZ may be an effective therapy for liver fibrosis without severe adverse effects.

## 1. Introduction

Liver cirrhosis, the end stage of chronic liver diseases, is caused by progressive fibrosis that ultimately results in nodular regeneration with loss of function [1,2]. Liver fibrosis is a common feature of chronic liver diseases regardless of etiology, which are characterized by the excessive accumulation of extracellular matrix components and activation of hepatic stellate cells (HSCs) [3,4].

HSCs are observed in the perisinusoidal space of the liver and are characterized by two distinct phenotypes. One is a quiescent phenotype in the normal liver, whereas the other is an activated phenotype in the fibrotic liver. Activated HSCs are thought to be major mediators of liver fibrosis, and their generation is maintained during liver fibrogenesis [5]. The epigenetic machinery, including DNA methylation, plays a crucial role in the transition of quiescent HSCs to activated HSCs [6,7,8]. It is expected that epigenetic therapy will be effective in changing the phenotype of HSCs from activated to quiescent and in inhibiting liver fibrogenesis [9,10].

Hydralazine (HDZ), an antihypertensive agent, also exhibits non-nucleoside demethylating activity [11,12]. As an epigenetic cancer therapy, HDZ-valproate is being repositioned as an oral DNA methyltransferase 1 and histone deacetylase inhibitor in cervical cancer, myelodysplastic syndrome, and cutaneous T-cell lymphoma [13,14,15]. It is noteworthy that HDZ-valproate is safe when used alone or in combination with chemotherapy or chemoradiation [16]. Recently, in addition to cancers, the therapeutic effects of HDZ-induced demethylation on chronic diseases, such as cardiac dysfunction and renal fibrosis, have been demonstrated [17,18]. However, the effect of HDZ on liver fibrosis remains unclear.

As an epigenetic therapy for liver fibrosis, the demethylation of HSCs using 5-aza-2′-deoxycytidine (5-aza-dC) can change their phenotype and exert an antifibrotic effect [19]. The treatment of HSCs with 5-aza-dC blocks their transdifferentiation and downregulates fibrotic genes, such as collagen type I alpha 1 (*Cola1*) [20]. 5-aza-dC also induces the expression of antifibrotic genes, such as RAS protein activator like 1 (*Rasal1*) and phosphatase and tensin homolog deleted from chromosome 10 (*Pten*), in HSCs [21,22]. However, in general, 5-aza-dC has been reported to induce excessive cytotoxicity at a high dose [23]. To obtain enhanced demethylation activity with lower cytotoxicity, combined treatment with another demethylating agent with 5-aza-dC is desirable.

In this study, we delivered a combined treatment with 5-aza-dC and HDZ to obtain an enhanced antifibrotic effect with lower cytotoxicity. Additionally, as a potential molecular mechanism underlying the sensitization to the antifibrotic effect of 5-aza-dC by HDZ, the expression levels of rat equilibrative nucleoside transporter genes (*rEnt1*, *rEnt2*, and *rEnt3*) were analyzed in this study.

## 2. Materials and Methods

### 2.1. Cell Culture and Reagents

A rat hepatic stellate cell line (HSC-T6) was cultured in Dulbecco’s Modified Eagle’s Medium supplemented with 10% fetal bovine serum (Both Invitrogen, Waltham, MA, USA) and 1% penicillin/streptomycin in an incubator at 37 °C, with exposure to a humidified atmosphere of 5% CO_2_.

### 2.2. Treatment of Cells with 5-Aza-dC and/or HDZ

5-Aza-dC and HDZ were purchased from Sigma Inc. (St. Louis, MO, USA). 5-Aza-dC and HDZ were dissolved in PBS and diluted in cell culture medium to obtain final concentrations, as specified. Cells were seeded overnight in culture dishes, and 5-aza-dC and/or HDZ were refreshed every 24 h until the 96 h of treatment ended.

### 2.3. WST-1 Assay

Cellular proliferation was measured using the WST-1 assay. Cells (5 × 10^3^) were seeded in 96-well plates and cultured at 37 °C in a humid chamber with 5% CO_2_. WST (5 mg/mL) was added to each well and incubated with the cells at 37 °C for 2 h. The optical density was measured at 450 nm. The percentage of viability was calculated according to the following formula: viability% = T/C × 100%, where T and C refer to the absorbance of the treatment group and the control, respectively. Data are presented as the mean ± SD of three independent experiments performed in triplicate.

### 2.4. Semi-Quantitative Real-Time Reverse Transcription PCR (RT–PCR)

Total RNA was extracted using an RNeasy^®^ Mini Kit (Qiagen, Hilden, Germany). cDNA was synthesized from 1 μg of total RNA using a High Capacity RNA to cDNA Master Mix (Thermo Fisher Scientific, Waltham, MA, USA). mRNA expression levels were measured by quantitative PCR using the StepOnePlus^TM^ Real-Time PCR^®^ (Thermo Fisher Scientific, Waltham, MA, USA). Peptidylprolyl isomerase A (*Ppia*) was used as an endogenous control, and the expression of the target genes was determined using the comparative threshold value. The primer sequences for *Ppia*, *Cola1*, αSMA, *Rasal1*, *Pten*, *rEnt1*, *rEnt2*, and *rEnt3* were as reported previously [20,22,24,25,26].

### 2.5. Statistical Analysis

The differences in the mean methylation levels were analyzed using Welch’s *t*-test (GraphPad Prism version 6.04 (GraphPad Software, La Jolla, CA, USA)). All the tests were two-tailed, and significance was set at *p* < 0.05.

## 3. Results

### 3.1. Toxicity of HDZ and 5-Aza-dC in HSC-T6 cells

A WST-1 assay was performed to assess the toxicity of HDZ at 10 and 50 μg/mL in HSC-T6 cells. These concentrations of hydralazine were determined with reference to the literature [18]. It was found that 10 μg/mL of hydralazine does not show any toxicity in HSC-T6 cells (Appendix A) As for 5-aza-dC, it was reported that 1 μM of 5-aza-dC does not show any toxicity in rat HSC cells [19]. Therefore, the concentrations of HDZ and 5-aza-dC were determined to 10 μg/mL and 1 μM, respectively.

### 3.2. Effect of Combined Treatment with 5-Aza-dC and HDZ on Cell Proliferation

A WST-1 assay was performed to determine the antiproliferative effect of combined treatment with 5-aza-dC and HDZ in HSC-T6 cells. The percentage cell viability with 5-aza-dC, HDZ, and combined treatment vs. a vehicle-only control was 101.4 ± 2.5, 95.2 ± 5.7, and 79.2 ± 0.7 (*p* < 0.01 for 5-aza-dC and *p* < 0.01 for HDZ), respectively, in the 48 h treatment, and 52.4 ± 5.6, 65.9 ± 3.4, and 29.9 ± 1.3 (*p* < 0.01 for 5-aza-dC and *p* < 0.01 for HDZ), respectively, in the 96 h treatment (Figure 1). The antiproliferative effect was more prominent in the combined treatment vs. the single treatments with 5-aza-dC and HDZ. HSC-T6 cells did not show any morphological changes before or after combined treatment.

### 3.3. Effect of the Combined Treatment with 5-Aza-dC and HDZ on Cola1, α-SMA, Rasal1, and Pten mRNA Expression in HSC-T6 Cells

To determine the antifibrotic effect of the combined treatment with 5-aza-dC and HDZ in HSC-T6 cells, the mRNA levels of fibrosis-related genes were measured. The expression level of *Cola1*, a fibrotic gene, was decreased after 5-aza-dC and the combined treatment, but not after HDZ treatment. On the other hand, the expression level of α-SMA, a fibrotic gene, was not decreased after any treatment, but was increased after 5-aza-dC and the combined treatment, unexpectedly (Figure 2). Accordingly, the expression levels of *Rasal1* and *Pten*, which are antifibrotic genes, were increased after 5-aza-dC and the combined treatment, but not after HDZ treatment (Figure 3). These results indicated that HDZ sensitizes to the antifibrotic effect of 5-aza-dC in HSC-T6 cells.

### 3.4. Effect of HDZ on the mRNA Expression of Three Rat Equilibrative Nucleoside Transporter Genes (rEnt1, rEnt2, and rEnt3) in HSC-T6 Cells

To determine whether HDZ upregulates nucleoside transporter genes and enhances 5-aza-dC activity, the mRNA levels of three rat equilibrative nucleoside transporter genes (*rEnt1*, *rEnt2*, and *rEnt3*) were measured. The expression level of *rEnt1* was increased in the 96 h treatment with 5-aza-dC and combined treatment but not in that with HDZ (Figure 4). The expression level of *rEnt2* seemed to decrease in the 48 h treatment with 5-aza-dC and combined treatment, and the expression level of *rEnt3* seemed to increase in the 96 h treatment with HDZ and combined treatment; however, these results were not statistically significant. These data suggested that HDZ does not affect the expression of nucleoside transporter genes in HSC-T6 cells.

## 4. Discussion

HSC activation is essential for the development of liver fibrosis. The DNA demethylation of HSCs using 5-aza-dC yielded antiproliferative and antifibrotic effects; however, in a clinical setting, the cytotoxicity of 5-aza-dC at a high dose is a serious problem. In this study, to enhance the antifibrotic effect using a low dose of 5-aza-dC, we performed a combined treatment with 5-aza-dC and HDZ in HSC-T6 cells. HDZ sensitized to the antifibrotic effect of 5-aza-dC in HSC-T6 cells.

Regarding the expression levels of *Cola1* and α-SMA, two fibrotic genes, *Cola1* expression was clearly decreased after 5-aza-dC and the combined treatment, but α-SMA expression was unexpectedly increased after 5-aza-dC and the combined treatment. *Cola1* expression might be more sensitive to 5-aza-dC treatment than α-SMA expression. However, the data should be interpreted with caution due to the limitations of mRNA expression analysis. Protein expression analysis will be desirable in future studies.

HDZ is a very well-known vasodilator that is used to treat high blood pressure. It also exhibits non-nucleoside demethylating activity. In the kidney, low dose HDZ prevents renal fibrosis through the demethylation of the *Rasal1* promoter in a murine model of acute kidney injury [18]. In our liver study, HDZ itself did not exert an antiproliferative effect or demethylation of *Rasal1* in HSC-T6 cells. The demethylation profile of genes might be different between the kidney and the liver.

Regarding the antihypertensive action of HDZ, in a mouse renal fibrosis model [18], the hydralazine dose for demethylating action, which is 10 to 40 times lower than that for antihypertensive action, does not lower blood pressure. Additionally, the demethylating dose of hydralazine is well tolerated in a clinical setting [16]. However, it is largely unknown whether the combined treatment with 5-aza-dC and hydralazine is safe and effective in a clinical setting. It is desirable that the antifibrotic effect of combined treatment and the mechanism of the drug’s action be studied in primary HSCs and in in vivo animal models.

5-aza-dC is a prodrug that requires metabolic activation by deoxycytidine kinase. The active form of 5-aza-dC, 5-aza-dCTP, incorporates into DNA and inhibits DNA methyltransferase. In humans, the half-life of 5-aza-dC is 15 to 25 min due to rapid inactivation by liver cytidine deaminase. The major side effect produced by 5-aza-dC is myelosuppression, in high doses [27]. Therefore, a low dose-schedule is adopted in clinical trials [23].

The molecular mechanism underlying the enhancement of the demethylating effect of 5-aza-dC by HDZ remains unknown. In a cervical cancer cell line, it was revealed that HDZ upregulates the human equilibrative nucleoside transporter 1 gene (*hEnt1*) and restores gemcitabine sensitivity [28]. It was also shown that high expression levels of *hEnt1* are closely associated with the response to 5-aza-dC in vitro and in vivo [29,30]. We hypothesized that HDZ upregulates nucleoside transporter genes and enhances the activity of 5-aza-dC. However, HDZ did not upregulate nucleoside transporter genes in HSC-T6 cells (Figure 4). The expression of other genes that enhance the action of 5-aza-dC might be affected by HDZ.

Further confirmation using primary HSCs and in vivo animal models is desirable, but considering that HDZ is used as an antihypertensive drug for many patients without adverse effects, combined treatment with 5-aza-dC and HDZ might be a promising demethylating treatment for liver fibrosis.

## Figures and Tables

**Figure 1 biology-09-00117-f001:**
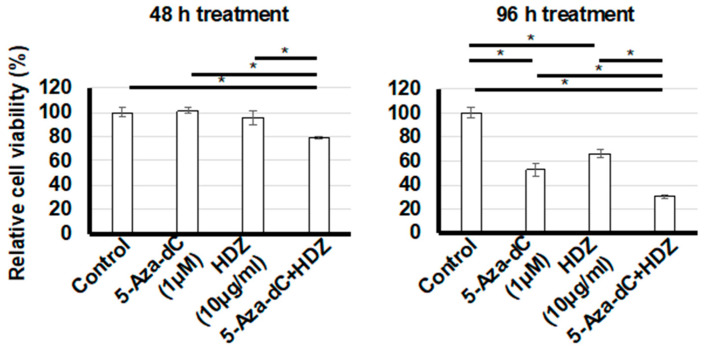
Proliferation of HSC-T6 cells (hepatic stellate cells) treated with 5-aza-2′-deoxycytidine (5-aza-dC), hydralazine (HDZ), and 5-aza-dC and HDZ combined. Cell proliferation was measured using the WST1 assay. The effects of 5-aza-dC, HDZ, and combined treatment with 5-aza-dC and HDZ were compared with untreated control cells. Data are the mean ± SD (n = 3). * *p* < 0.01 vs. the untreated control.

**Figure 2 biology-09-00117-f002:**
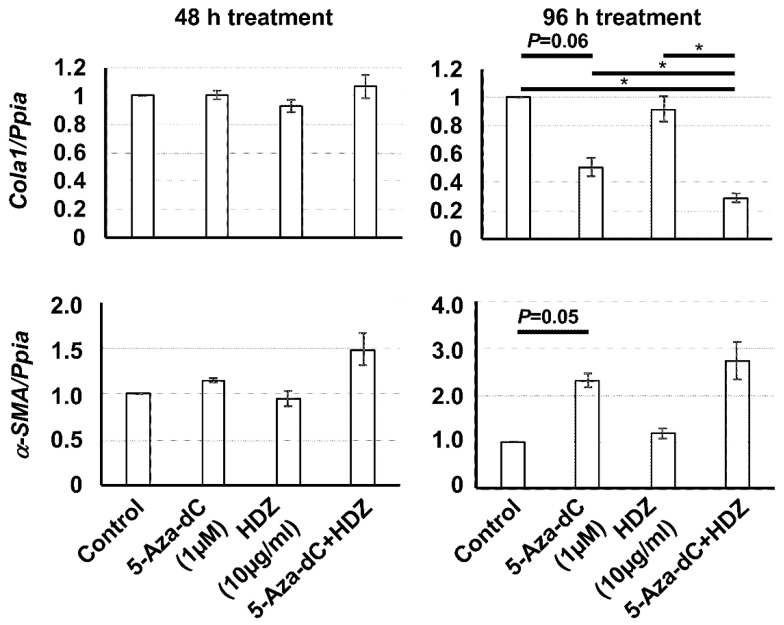
The mRNA expression levels of *Cola1* and aSMA, fibrotic genes, in HSC-T6 cells treated with 5-aza-dC, HDZ, and 5-aza-dC and HDZ combined. The mRNA expression levels were measured using semi-quantitative RT–PCR. The effects of 5-aza-dC, HDZ, and combined treatment with 5-aza-dC and HDZ were compared with untreated control cells. Data are the mean ± SD (n = 3). * *p* < 0.05 vs. the untreated control.

**Figure 3 biology-09-00117-f003:**
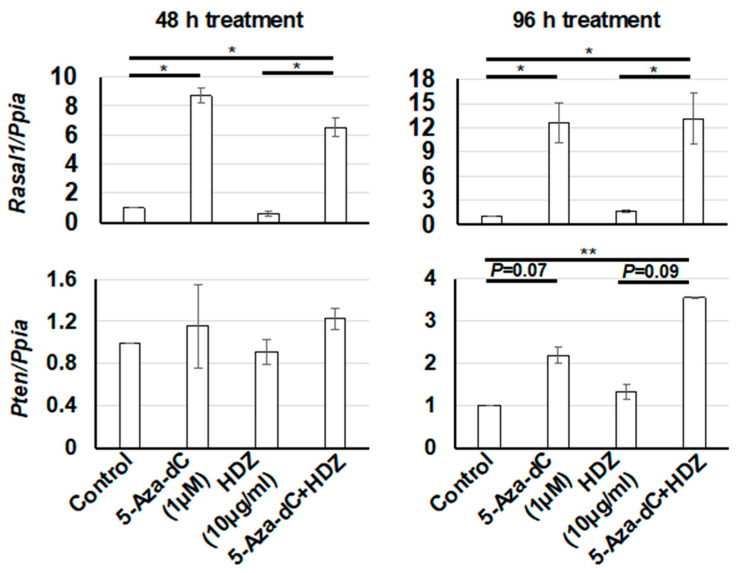
mRNA expression levels of *Rasal1* and *Pten*, which are antifibrotic genes, in HSC-T6 cells treated with 5-aza-dC, HDZ, and 5-aza-dC and HDZ combined. The mRNA expression levels were measured by semi-quantitative RT–PCR. The effects of 5-aza-dC, HDZ, and combined treatment with 5-aza-dC and HDZ were compared with untreated control cells. Data are the mean ± SD (n = 3). * *p* < 0.05 and ** *p* < 0.01 vs. the untreated control.

**Figure 4 biology-09-00117-f004:**
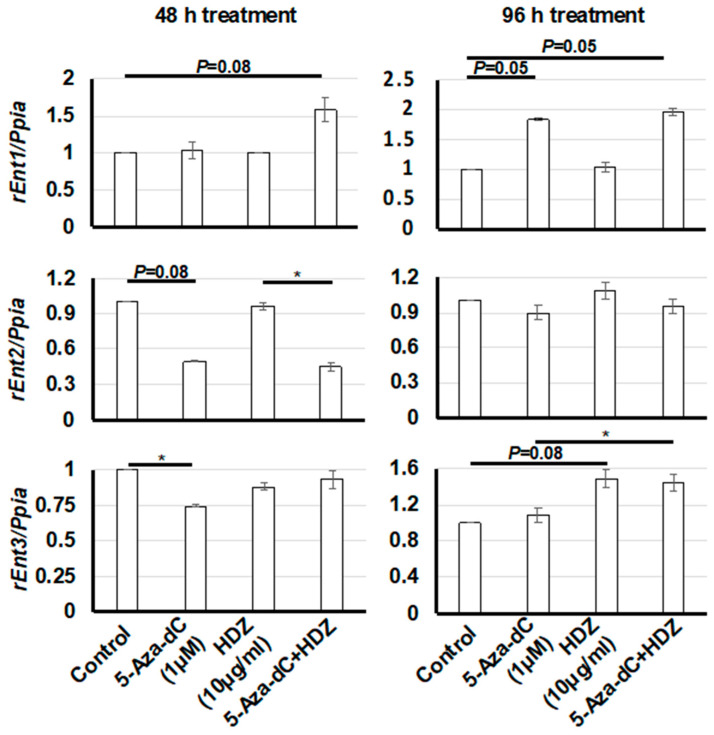
mRNA expression levels of *rEnt1*, *rEnt2*, and *rEnt3*, which are rat equilibrative nucleoside transporter genes, in HSC-T6 cells treated with 5-aza-dC, HDZ, and 5-aza-dC and HDZ combined. The mRNA expression levels were measured by semi-quantitative RT–PCR. The effects of 5-aza-dC, HDZ, and combined treatment with 5-aza-dC and HDZ were compared with untreated control cells. Data are the mean ± SD (n = 3). * *p* < 0.05 vs. the untreated control.

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
