# Peer review of "Hydralazine Sensitizes to the Antifibrotic Effect of 5-Aza-2′-deoxycytidine in Hepatic Stellate Cells"

_biology, 2020, doi:10.3390/biology9060117_

Round 1

Reviewer 1 Report

In this study, the authors use the nucleoside drug 5– AZA DC in combination with the antihypertensive hydralazine to inhibit liver fibrosis for the treatment of end-stage liver failure. They use an epigenetic approach of demethylation of fibrotic genes. HDC is known to act as a demethylation agent in experiments with cervical cancer and 5– AZA DC also has some epigenetic effects, however, with toxicity at high doses. The evidence from this data is that some fibrotic genes can be demethylated by this combined treatment. However, very little investigation has been conducted on the toxicity of these drugs in the cell culture setting. I would like to get the authors to consider conducting experiments that will determine the therapeutic indices when compared with on drug cells. This can be done in the cell culture setting or with clonogenic assays. 5– AZA DC is a ribonucleotide reductase drug and not much is mentioned about the impact of this drug on the nucleotide pool and its independent effects. If this combination was to be a therapy would, the patients administered with HDC have low blood pressure? Finally, no clear mechanism of the drug action emerges out of this study.

Reviewer 2 Report

Line 1-3: please rephrase the sentence because it is wordy.

Line 4-7: I suppose you could remove all the etiologies, because the term "regardless of etiology" is used.

Line 10-11: it seems a subject is missing for the verb "are".

Line 17-18: What software was used to perform analyses? Which were the reasons to employ Welch's test? P-value was two-tailed? Please, specify.

Reviewer 3 Report

The study describes the effect of the combined treatment of 5-aza-dC and hydralazine in rat liver stellate cell line (HSC-T6). The authors found that Hydralazine enhances the sensitizing antifibrotic activity of 5-aza-dC.

Authors should consider the following comments:

Results:

  1. Have authors observed any changes in cell morphology before and after combined treatment? Provide a brief description of these changes.
  2. The authors provided a very limited study of the expression of mRNA markers. There is no data on the level of mRNA expression of classical markers such as alpha-SMA, PDGF, vimentin. Such data should be added.
  3. The results of the mRNA expression of the studied genes, as far as possible, should be confirmed by protein expression for the same genes.

Discussion:

  1. Line 20-21 “The expression of other genes…” – Could you write this phrase more clearly?

Additional points

Line 50 - characterized be excessive - must be “characterized by excessive”

Line 51 - hepatic stellated (stellate) cells – must be “hepatic stellate cells”

Conclusion:

Primary HSC cells are desirable for complete confirmation of the obtained results of antifibrotic effects in the combined treatment of 5-aza-dC and HDZ

Round 2

Reviewer 1 Report

The authors have addressed my first query with respect to toxicity. However, two other questions have not been answered which is unacceptable.

Reviewer 3 Report

Dear authors,

I don't see any corrections in the new version of your manuscript. Maybe you added the old version of the manuscript by an error?

"The results of the mRNA expression of the studied genes, as far as possible, should be confirmed by protein expression for the same genes. 
Authors’ reply: We think this comment is important. However, methylation directly regulate mRNA expression, and mRNA expression is not always correlated with protein expression since post-transcriptional processes are involved in the protein synthesis. So, we analyze mRNA expression in this study."

I dоn't completely agree with your explanation of not showing the results of protein expression where you state: "mRNA expression does not always correlate with protein expression."

Liver fibrosis is primarily the accumulation of extracellular matrix proteins. The role of HSC cells in this process is very important, therefore, it is necessary to show and discuss the levels of protein markers of fibrosis before and after treatment.

Round 3

Reviewer 3 Report

The authors answered all my comments.